# Sistas Taking a Stand for Breast Cancer Research (STAR) Study: A Community-Based Participatory Genetic Research Study to Enhance Participation and Breast Cancer Equity among African American Women in Memphis, TN

**DOI:** 10.3390/ijerph15122899

**Published:** 2018-12-18

**Authors:** Alana Smith, Gregory A. Vidal, Elizabeth Pritchard, Ryan Blue, Michelle Y. Martin, LaShanta J. Rice, Gwendolynn Brown, Athena Starlard-Davenport

**Affiliations:** 1Department of Genetics, Genomics and Informatics, University of Tennessee Health Science Center, Memphis, TN 38163, USA; aantoin1@uthsc.edu; 2Department of Medicine, The University of Tennessee West Cancer Center, Memphis, TN 38163, USA; gvidal1@uthsc.edu (G.A.V.); fpritcha@uthsc.edu (E.P.); 3Division of Hematology and Oncology, Department of Medicine, University of Tennessee Health Science Center, 7945 Wolf River Boulevard, Memphis, TN 38138, USA; 4College of Nursing, University of Tennessee Health Science Center, Memphis, TN 38163, USA; rblue2@uthsc.edu; 5Department of Preventive Medicine, University of Tennessee Health Science Center, Memphis, TN 38163, USA; mmart126@uthsc.edu; 6School of Health Sciences, Online Learning, Stratford University, 3201 Jermantown Road, Ste 500, Fairfax, VA 22030, USA; lashantajrice@gmail.com; 7Carin and Sharin Breast Cancer Support Group, Memphis, TN 38613, USA; gwendolynnbro@gmail.com

**Keywords:** African American women, community-based participatory research, genetics, breast cancer, health equity, Memphis, TN

## Abstract

African American women are substantially underrepresented in breast cancer genetic research studies and clinical trials, yet they are more likely to die from breast cancer. Lack of trust in the medical community is a major barrier preventing the successful recruitment of African Americans into research studies. When considering the city of Memphis, TN, where the percentage of African Americans is significantly higher than the national average and it has a high rate of breast cancer mortality inequities among African American women, we evaluated the feasibility of utilizing a community-based participatory (CBPR) approach for recruiting African American women into a breast cancer genetic study, called the Sistas Taking A Stand for Breast Cancer Research (STAR) study. From June 2016 and December 2017, African American women age 18 and above were recruited to provide a 2 mL saliva specimen and complete a health questionnaire. A total of 364 African American women provided a saliva sample and completed the health questionnaire. Greater than 85% agreed to be contacted for future studies. Educational workshops on the importance of participating in cancer genetic research studies, followed by question and answer sessions, were most successful in recruitment. Overall, the participants expressed a strong interest and a willingness to participate in the STAR study. Our findings highlight the importance of implementing a CBPR approach that provides an educational component detailing the importance of participating in cancer genetic research studies and that includes prominent community advocates to build trust within the community.

## 1. Introduction

Breast cancer is a major health concern in the United States. This year alone, more than 260,000 women will be diagnosed with breast cancer in the United States (US) and more than 40,000 of those women will die from the disease [1]. Although, historically, non-Hispanic white women have been most at risk of developing breast cancer, African American women are now equally as likely to develop breast cancer as compared to non-Hispanic white women [2]. African American women also have a significantly higher risk of dying from breast cancer than their white counterparts. In 2017, breast cancer death rates were reportedly 39% higher in African American women as compared to non-Hispanic white women [1]. Thus, as more African American women are diagnosed with breast cancer, it is expected that more of these women will die from the disease.

Multiple risk factors across the breast cancer spectrum have been identified to explain the racial disparity in breast cancer mortality, including demographic and biological factors [3,4,5,6]. For instance, African American women are most likely to be diagnosed with aggressive forms of breast tumors that are hormone receptor-negative and that are associated with poorer breast cancer survival outcomes when compared to all other ethnic groups. This suggests that genetic differences in breast tumor biology may contribute to the breast cancer mortality disparity gap between African American and non-Hispanic white women [4,5,6,7]. However, the exact causes for these racial breast cancer inequities in mortality rates between African American and non-Hispanic white women is not fully understood.

Despite an increase in the number of genetic research studies examining the biological determinants of breast cancer mortality disparities in African American women [8,9,10,11,12,13], African Americans particularly those residing in southern regions of the United States, are substantially underrepresented in genetic research studies [14,15,16,17] and clinical trials [18,19,20]. African Americans represent approximately 13% of the United States population [21], yet, according to the National Institutes of Health (NIH) National Institute on Minority Health and Health Disparities, less than 10% of African Americans are enrolled in clinical trials and few participate in genetic research studies [22,23,24]. Several factors, including stigma, lack of trust in academic researchers and clinicians and researchers’ lack of awareness and knowledge about cultural differences, have been attributed to barriers in recruitment of racial minority groups into health-related research studies [23,25,26]. Recruitment models that leverage community-based participatory research (CBPR) capacity to educate volunteers about the importance of participating in genetic research studies, while addressing the historical issues of fear and mistrust in the medical community may prove beneficial in reducing inequalities in minority participation in cancer genetic research studies and health inequities [27]. However, literature that is specifically focused on best practices for recruitment and retention of African American women for cancer genetic research studies is limited.

The city of Memphis, TN, where the percentage of African Americans is significantly higher than the national average, has a significantly high rate of breast cancer mortality disparities among African American women [28,29,30]. In 2016, a community-based breast cancer genetic research study, called the Sistas Taking A Stand for breast cancer genetic Research (STAR) study, was developed by the Principal Investigator, Dr. Athena Starlard-Davenport, and her research team at the University of Tennessee Health Science Center in Memphis, TN to investigate the biological causes of breast cancer inequities among African American women in Memphis. Specifically, the objectives of the STAR study are to: (1) identify genetic modifiers of breast cancer risk and recurrence in African American women, (2) identify environmental and lifestyle factors (e.g., obesity, body mass index (BMI), smoking status, alcohol consumption) that influence breast cancer risk and survival outcomes, and (3) educate women about breast cancer prevention strategies and the importance of participating in cancer genetic research studies.

In order to better understand the biological determinants of breast cancer mortality inequities among African American women in Memphis, we hypothesized that establishing partnerships with prominent community breast cancer advocates and educating the Memphis community on the importance of participating in our cancer genetic research study would promote the successful recruitment and retainment of African American women in our STAR study. The objective of this study was to evaluate the feasibility of utilizing a CBPR approach for recruiting and retaining African American women in Memphis in the STAR study. In this paper, we describe how we utilized a CBPR approach to recruit African American women for our STAR breast cancer genetics research study.

## 2. Materials and Methods

### 2.1. Research Team

The STAR study is a community-based participatory genetic research study that was initiated in the autumn of 2016. The primary objective of the STAR study is to educate, recruit, and retain African American women participants throughout the Memphis community in our cancer genetic research study. Efforts to recruit participants in the STAR study involved a group of racially diverse team members who had a strong presence in the Memphis community. Team members were identified through word of mouth, interviews on local news coverage, and through interactions at the monthly Memphis Breast Cancer Consortium-Common Table Health Alliance meetings. Specifically, team members included academic professionals at the University of Tennessee Health Science Center and Southwest Tennessee Community College with cancer research experience, breast oncologists, and a research oncology nurse at the West Cancer Center, and a prominent leader and founder of a local breast cancer support group who was instrumental in providing suggestions and comments on the study logo, culturally sensitive educational materials, and strategies to recruit African American women in the STAR study. The research team also included non-paid undergraduate and graduate student volunteers. All team members completed study specific training that addressed consent procedures, saliva sample collection, quality control procedures, confidentiality, and data security protection. All team members also completed human subjects’ protection training that is required by the University of Tennessee Health Science Center Institutional Review Board. Written informed consent was obtained from all participants before they participated in the STAR study. Team members did not participate in the STAR study. The study was conducted in accordance with the Declaration of Helsinki, and the protocol was approved by the Institutional Review Board of the University of Tennessee Health Science Center, Memphis, TN. IRB protocol 16-04551-XP was approved to recruit and consent volunteers at community-based outreach events, and IRB protocol 16-04502-XP was used to recruit and consent breast cancer patients that were being treated at the West Cancer Center.

### 2.2. Eligibility Criteria

Women who self-reported as being African American aged 18 years or older and who were able to: (1) provide written informed consent, (2) complete a three-page health questionnaire, and (3) provide a 2-mL saliva specimen were eligible to enroll in the STAR study.

### 2.3. Study Design

The study and consent process was explained to the volunteer prior to enrollment. STAR study staff explained to volunteers that they would not be provided with personalized research results. Women who indicated an interest in participating were further asked to read and sign the IRB approved written consent form. The consent form also contained a brief introduction and basic description of breast cancer genetic research. During the consent process, the volunteer was also given the opportunity to indicate their willingness to follow up contact within two years to be informed of research opportunities and to ask about any new developments in their health. Participants who agreed to follow up contact provided their name, mailing address, e-mail address, and/or phone number on their consent form. Participants were not excluded from the study if they were not willing to be contacted in the future.

After the participants provided written consent, participants were provided with a confidential three-page self-report questionnaire. The questionnaire consisted of demographics, which included information on participants’ reproductive history, diet and lifestyle factors, family history of cancer, and history of genetic testing.

After participants provided written consent and completed the health questionnaire, participants were asked to provide a 2-mL saliva sample. The 2 mL saliva sample was collected using Oragene^®^•OG-500 DNA Self-Collection Kit (DNA Genotek, Ottawa, Ontario, Canada) and was stored in a locked cabinet at room temperature. The average time for participants to obtain written informed consent, complete the health questionnaire, and collect a saliva specimen was approximately 8–10 min.

### 2.4. Recruitment Sites

Establishment of collaborations and partnerships with key leaders who have a strong presence in the Memphis community was critical to engage volunteers and educate them about the importance of participating in the STAR study. Volunteers were recruited at several venues and events in Memphis, TN between June 2016 and December 2017. The venues and events included the West Cancer Center, Sista Strut 3K Breast Cancer Walk annual event, Benjamin L. Hooks Central Library, and ‘Carin and Sharin’ breast cancer support group. Participants at the Breast Cancer Walk received a bottle of water with the STAR logo on it soon after providing a saliva specimen and completing the health questionnaire. The recruitment and consenting of breast cancer patients who were being treated at the West Cancer Center was conducted in close collaboration with breast oncologists and a research nurse at the West Cancer Center in Memphis, TN.

### 2.5. Educational Workshops

Workshops to engage and educate volunteers on the importance of participating in cancer genetic research were presented at the Live! Memphis Annual Breast Cancer Summit, Southwest Tennessee Community College, and the monthly Carin and Sharin breast cancer support group meeting. The Carin and Sharin breast cancer support group, led by Mrs. Gwendolyn Brown, is an organization that was developed to address the breast cancer mortality disparities among socio-economically disadvantaged, inner city, African American women in Memphis, TN [29,30]. Culturally appropriate brochures and flyers with a specific mission to reduce breast cancer disparities throughout the Memphis community were distributed to volunteers at all workshops and venues. All educational workshops were immediately followed by a 10–15-min question and answering (Q & A) session. Immediately following the Q & A session, volunteers were given the opportunity to voluntarily donate a saliva specimen and complete the health questionnaire.

By contrast, due to the nature of the event, educational workshops were not provided to volunteers at the Sista Strut 3K Breast Cancer Walk or patients being treated for breast cancer at the West Cancer Center.

### 2.6. Statistical Analysis

A total of 364 African American women participated in the STAR study by providing a saliva sample and completing the health questionnaire. Volunteers were recruited and consented at several venues and events in Memphis, TN between June 2016 and December 2017. Records of participants’ consent forms that provided the option to be contacted after two years were systematically collected by the study PI and were used to calculate the follow-up recruitment rates. Demographic characteristics and selected risk factors for breast cancer were compared between cases and controls using T-tests for continuous variables and Chi-square tests for categorical data. Variables were dichotomized in which women who reported any alcohol use or tobacco use were assigned a value of “1” and women who did not report consuming alcohol, were assigned a “0”. Data were expressed as the mean ± standard deviation. All *p*-values were two-sided and considered significant at the alpha 0.05 level. All statistical analyses were performed using GraphPad Prism, version 7 software (La Jolla, CA, USA).

## 3. Results

### 3.1. Target Recruitment Goal Met

Our initial goal was to recruit a total of 250 African American women for the STAR study between June 2016 and December 2017. We exceeded our goal by recruiting a total of 364 African American women who provided both a saliva sample and completed the health questionnaire. The total number of participants consisted of 94 breast cancer cases, including breast cancer survivors, and 270 healthy women controls (Table 1). Women who participated in the STAR study were at least age 18 or older.

### 3.2. Demographic and Clinical Characteristics

A table summarizing the demographic characteristics of the STAR study participants is shown in Table 1. The mean ± SD age of breast cancer cases and the comparison group was 56.7 ± 13.6 and 48.2 ± 13.7 years, respectively. The minimum and maximum age among the breast cancer cases was 20 and 91 years. The minimum and maximum age among healthy volunteers was 18 and 79 years, respectively. As shown in Table 1, the age of breast cancer patients was significantly higher than the control group on age (*p* < 0.0001).

We also assessed whether behaviors and lifestyle factors that are known to increase the risk of breast cancer development, specifically obesity, weekly alcohol consumption, and tobacco use ever were higher in women with breast cancer as compared to controls. To determine the relationship between obesity, alcohol consumption, or tobacco use and breast cancer status, we included the following questions in the health questionnaire: “How tall are you?”, What is your weight?”, “Do you smoke cigarettes or vapor?”, “How often do you smoke cigarettes or vapor?”, and “How many alcoholic drinks (beer, wine, liquor) do you currently drink weekly?” Interestingly, the percentage of women who consumed alcohol was significantly higher among controls versus breast cancer cases. Although obesity was a leading comorbidity among all participants, regardless of disease state, obesity status was not significantly different between cases and controls. Furthermore, less than 10% of cases and 25% of controls used tobacco. There was no significant difference between cases and controls for menopausal status and family history of cancer.

### 3.3. Evaluation of Our CBPR Recruitment Strategy

Between June 2016 and December 2017, we consented 364 study participants at a total of six community outreach Memphis events and venues. Our most successful recruitment events were those where we provided both educational resources and workshops on the importance of participating in cancer genetics research (Figure 1). Specifically, these workshops were provided at the 2017 Live! Memphis Annual Breast Cancer Summit and breast cancer support group meetings, where the study volunteers listened to a 30-min seminar on the importance of participation in breast cancer genetics research, followed by a Q & A session. We consented a total of 205 women, of which 112 were recruited at the Live! Memphis Annual Breast Cancer Summit and 93 were recruited at the breast cancer support group meeting. The 2017 Annual Sista Strut 3K Breast Cancer Walk yielded a total of 132 African American women who provided written consent to participate in the STAR study. Only 27 breast cancer patients who were being treated for breast cancer at the breast oncology clinic provided written consent to participate in the STAR study.

More than 88% of STAR study participants provided written consent and contact information (i.e., email, phone number, home address) to be contacted in two years by STAR study support staff to obtain any new information about their health and new research opportunities (Figure 2). Approximately 85% of cases (*n* = 80) consented to follow up contact in two years. Similarly, 89% of healthy (control) participants (*n* = 241) agreed to follow up contact. Among breast cancer cases, 90% of participants who attended the educational workshops provided written consent to be follow up contacted. Similarly, close to 90% of cases and controls who participated at the Sista Strut 5K Walk or were treated for breast cancer at the West Cancer Center (clinic) provided written consent to follow up contact.

### 3.4. Participants’ Attitudes about Genetic Research

Although participants understood that they would not receive any research results, most participants expressed a strong interest and a willingness to participate in the STAR study, as previously described [31]. Most participants made comments such as: “Your research will help future generations!” Other participants expressed a personal reason for donating particularly noting that loss of a family member or friend to cancer motivated them to participate in our study. Additionally, several African American women in Memphis reported that they were less likely to be in research studies because they were never asked to participate [32].

## 4. Discussion

In this study, we evaluated the feasibility of utilizing a CBPR approach for recruiting and obtaining a saliva sample from African American women in Memphis for our breast cancer genetic research study: the STAR study. We exceeded our target goal of 250 participants during the recruitment period by recruiting 364 participants who provided a saliva specimen and completed a health questionnaire. Approximately 90% of the women in our study agreed to follow up contact to obtain information about their health.

It is well documented that a lack of trust in the medical community is a major reason why African Americans decline to participate in research studies [15,23,24,33,34,35,36]. Refusal to participate in research studies and clinical trials stems from historical evidence of racial injustices through the mistreatment of African American men in the U.S. Public Health Service Syphilis Study at Tuskegee [37,38]. In that study, rural African American men in Alabama were withheld treatment for syphilis and information about the disease. Additionally, the story of Mrs. Henrietta Lacks, an African American woman whose cervical cancer cells were taken and were commercially developed into the first immortalized cell line (HeLa cells) without her or her family’s knowledge, further raised concerns about privacy and patients’ rights within the African American community [39,40,41]. More recently, a study by Halbert et al. found that recruiting African-American women into a hereditary genetic counseling research study proved challenging primarily due to difficulty establishing contact with potential participants [42]. However, our findings show that African American women in Memphis are willing and wanting to participate in breast cancer genetic research studies.

Two major factors that contributed to the successful recruitment of African American women in the STAR study included: (1) the development of partnerships with prominent breast cancer advocates who had a strong presence in the Memphis community and (2) incorporation of culturally appropriate educational workshops on the importance of participating in breast cancer genetic research studies. Approaches that use a CBPR approach have yielded similar results to our research findings [27,43,44]. In a study by Ochs-Balcom et al., they found that a CBPR approach that included community members in the recruitment process and that provided clear communication strategies about the underlying benefit to potential volunteers allowed them to successfully recruit 341 African American women for their breast cancer genetics study [43]. A similar study by McElfish et al., who developed a CBPR partnership with a local clinic, was established to recruit and obtain saliva from an underserved minority Marshallese Pacific Islander community in Arkansas, was also established [27,44]. Their partnership with the Marshallese community yielded a recruitment rate of 95.5% and 96.6% that agreed to be contacted for future studies. The strength of their partnership was that the community had a pre-existing relationship with study staff who were bilingual Marshallese staff and who took time to fully describe the rationale for the genetics study to participants [27,44]. Other studies have also observed that community outreach initiatives that included an educational component on how their biospecimens would be used and the purpose for the research study further resulted in a high level of interest in biospecimen donation [45,46,47,48,49]. Collectively, our study, along with these and other studies, provide strong evidence that building trust within the community by developing partnerships with prominent advocates and enhancing the knowledge on the importance of participating in genetic research studies is critical for successful recruitment [27,43,44,50].

Our study has limitations. We were unable to determine how many women had as an initial interest in participating in our study due to limited study staff. Therefore, we were unable to calculate the recruitment rates. This was especially evident at the Sista Strut event that had over 5000 women in attendance. However, we were able to determine the percentage of participants who agreed to follow up contact. Additionally, our CBPR approach was limited to African American women who attended our recruitment events throughout the Memphis community. However, based on previous studies, it is plausible that our CBPR approach would yield similar recruitment results to other populations and underserved communities that may be difficult to recruit from.

A major advantage of our study is the population under study. African American women with breast cancer in Memphis, TN suffer disproportionately from higher breast cancer mortality rates when compared to their non-Hispanic white counterparts [28,29,30]. In fact, Memphis, TN has one of the highest breast cancer mortality disparity gaps between African American and non-Hispanic white women as compared to 49 of the largest US cities [29,30]. Thus, CBPR recruitment efforts to increase participation in breast cancer genetic research studies among African American women in Memphis is essential to achieve breast cancer health equity.

CBPR efforts to enhance participation in cancer genetics research have the potential to achieve breast cancer health equity. CBPR efforts can improve health outcomes in breast cancer screening, incidence, mortality, survivorship, and treatment of breast cancer [51,52]. The development of CBPR partnerships within the community can help in several ways: (1) it can inform the community of studies being conducted, (2) it allows for volunteers to gain additional knowledge of the disease being studied, and (3) it provides community members an opportunity to make a difference in their communities by participating in the research effort. Additionally, partnerships developed through CBPR efforts can help to guide researchers’ efforts in developing culturally tailored interventions to achieve breast cancer health equity by reducing breast cancer mortality disparity gaps between racial and ethnic groups [53,54,55].

## 5. Conclusions

In conclusion, our findings highlight the importance of implementing a CBPR approach that includes key community leadership needed to build trust with the local community and that provides educational resources on the importance of participating in cancer genetic research studies. Establishment of collaborations and partnerships with breast cancer advocates and key leaders in the Memphis community was essential for engaging, educating, and successfully recruiting participants for the STAR study. Workshops that educate African American women about strategies to prevent breast cancer development, combined with increasing their knowledge of the importance of participating in breast cancer research studies, was most successful for recruiting African American women in the STAR study. Overall, African American women in Memphis, TN are willing and wanting to participate in cancer genetics research studies. When considering the long history of racial inequality and discrimination that has marred Memphis for many years, future research efforts that incorporate culturally sensitive educational tools and interventions that will engage and support the African American community should also be considered when recruiting for genetic research studies.

## Figures and Tables

**Figure 1 ijerph-15-02899-f001:**
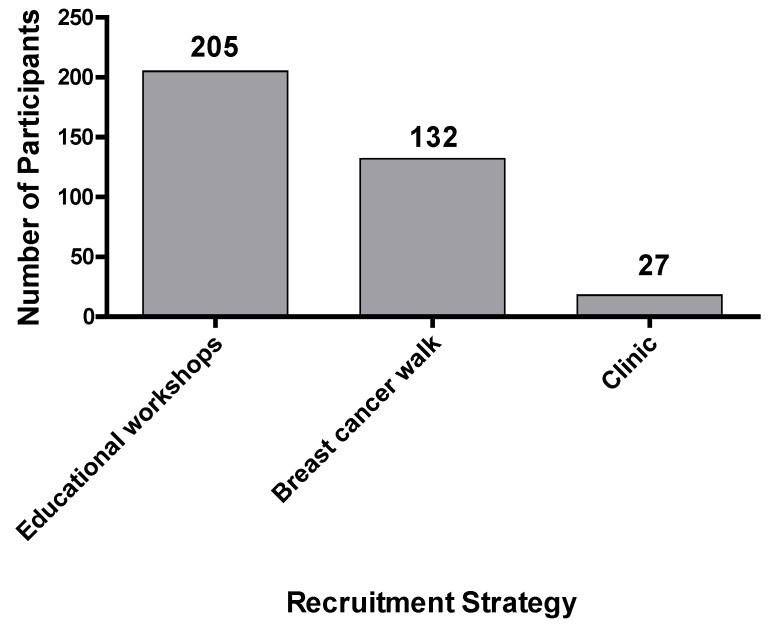
Strategies to recruit African American women in the Stand for breast cancer genetic Research (STAR) study.

**Figure 2 ijerph-15-02899-f002:**
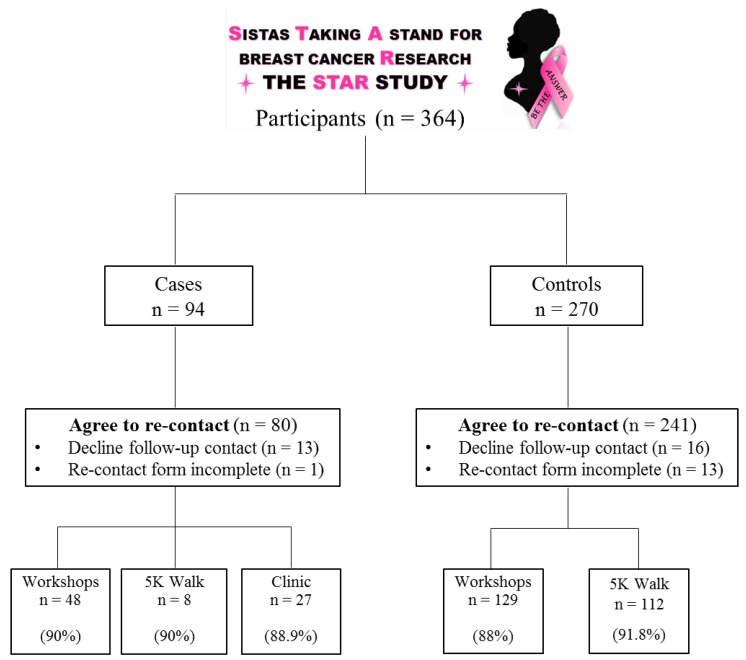
Flow chart of STAR study participant follow-up. The flow chart shows the number (*n*) of participants with breast cancer (cases) and without breast cancer (controls) who provided written consent to be re-contacted after two years from initial consent to participate in the STAR study Percentage (%) represents the percent of participants who agreed to follow-up contact to those who did not grant such permission. Data is reported for each of the recruitment strategies: workshops, 5K walk, or in the clinic.

**Table 1 ijerph-15-02899-t001:** Demographic and characteristics of study participants.

Characteristic	Cases (*n* = 94)	Controls (*n* = 270)	*p*-value
Age in Years, Mean ± SD	56.7 ± 13.6	48.2 ± 13.7	<0.0001
Minimum	20.0	18.0	
25% Percentile	46.5	38.0	
Median	55.0	50.0	
75% Percentile	67.0	59.0	
Maximum	91.0	79.0	
Menopause Status, *n* (%)			0.216
Pre-menopausal	39 (41.5%)	132 (48.9%)	
Post-menopausal	55 (58.5%)	138 (51.1%)	
Family History of Cancer, *n* (%)			0.574
Yes	53 (56.4%)	141 (52.2%)	
No	40 (42.6%)	122 (45.2%)	
Missing	1	5	
Comorbidity			
Obesity	46 (43.8%)	146 (54.1%)	0.390
Alcohol consumption	26 (27.7%)	141 (52.2%)	<0.0001
Tobacco use	5 (5.32%)	23 (8.52%)	0.316

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
