# Peer review of "Sistas Taking a Stand for Breast Cancer Research (STAR) Study: A Community-Based Participatory Genetic Research Study to Enhance Participation and Breast Cancer Equity among African American Women in Memphis, TN"

_ijerph, 2018, doi:10.3390/ijerph15122899_

Round 1

Reviewer 1 Report

Due Dec 10, 2018

Review request: International Journal of Environmental Research and Public Health

Manuscript ID: ijerph-402828

Title:      A Community-Based Participatory Approach to Enhance Participation Among African American Women in a Breast Cancer Genetics Research Study in Memphis, TN

Synopsis

To overcome the low participation of African American women in breast cancer genetic research studies due to the lack of their trust of the medical community, the Sistas Taking A Stand for Breast Cancer Research (STAR) study took a community-based participatory approach in the predominantly African American community in Memphis, Tennessee. A total of 364 African American women were recruited to provide a 2 mL saliva specimen and complete a health questionnaire through community events and clinics. Greater than 85% of the participants agreed to be contacted for future studies. Educational workshops followed by a question and answer session with prominent community advocates were the most successful strategy to build trust in recruitment.

Reviewer's conflict of interest: None

Comment to authors

Because the format of qualitative research reports allows flexibility, this reviewer's recommendations may become a matter of personal preferences and differences in opinion. In general, four criteria for evaluating qualitative research rigor are: 1) credibility, 2) dependability, 3) confirmability, and 4) transferability (Lincoln and Guba, 1985). That being said, the revision recommendations are listed below:

1.       Title. The tile did not use the abbreviation, the STAR. This may be a minor issue; however, authors may know that the STAR project in Tennessee was a known educational class-size study in 1999 for grade K-3 students. The 1999 STAR project, Student/Teacher Achievement Ratio study, happened in Tennessee and had an educational component. I wonder if a less confusing abbreviation was considered. Some people may get confused by the abbreviation. Line 6: The author "Michelle MartinF" contains a typo.

2.       Abstract. Line 26: "predominantly" is understandable; however, the percent African American in Memphis, TN, relative to the percent African American in the U.S. population may be more informative. Please clarify whether the 364 participants were either the total number of consented participants in the study or the total number of paired samples (saliva and questionnaire). Because this manuscript does not report the results of the genetic analysis, I made an assumption that this report is a part of the STAR study, or a different study to test recruitment effectiveness for the STAR study by the CBPR recruitment process.  

3.       Introduction: The needs and rational for the study and its design are well presented.

4.       Material and Methods: Please explain the difference between the protocols #16-04551-XP and #16-04502-XP, and who was consented to which study. If the study collected data from the group of ethnically diverse team members and academic professionals, they became research participants. Are they research participants, and was the plan to hire (as team members) a part of the CBPR protocol? These detailed questions become important when other organizations would like to replicate the findings of your study. How did you identify influential community members and how many stakeholder meetings were conducted? Did meeting minutes become  CBPR study data? Section 2.2 (Line 117) indicates the eligibility of human subjects whose saliva and questionnaire data were collected. This implies community members were not considered research subjects. (which is okay).

5.       2.3 Study Design; 2.4 Sites. One of the words causing confusion throughout this manuscript is "participant". I highly recommend using "volunteer" for potential participants at the information session, educational workshop, and clinics, and use "participant" for those who signed the consent form to be in the study (potentially two studies in this manuscript because there are two protocols). For instance, "participant" in line 122-131 could be "volunteer," and "participant" below line 132 could be a consented research subject. As described in limitations in the Discussion section, in the CBPR study, it could be 1000 attendees to a community event, 100 volunteers (equals possible research subjects) who were reached out to in some way, and 10 consented participants. A flow chart like Figure 2 is good and a similar chart can be created for recruitment. Figure 2 is a breakdown of the end result, not a recruitment diagram.

6.       2.4 Workshops; 2.5 Data Analysis. Again, please clarify how many consents, and when volunteers were consented.  

7.       3.1 Target; 3.2 Characteristics. The final result was shown like a case-control descriptive summary in Table 1. This is fine.

8.       3.3 evaluation; 3.4 attitude. By referencing Stringer (Action Research, Sage Publication 2007, p27), 1) the community-input processes can be better described, i.e. it still has the impression of a top-down decision to integrate the educational workshop for recruitment, 2) community mobilization process such as extra IRB-approved informational flyers during already existing community events can be better explained, 3) tactics for empowering key community members by letting them perform significant tasks need to be described, and 4) how the project emphasized working individually can be explained as well.

9.       Discussion. This section was well-written including the limitation of keeping track of all attendees at each event. If the research activities did not have a protocol-specific method to collect data on those numbers,  local news media can be used to estimate such data.

Overall, the project is well-designed and executed. Some confusion in the report needs to be clarified, including the existence of two protocols, the timing of consenting, and the definition of "participant". I recommend the authors consider what kind of information would help other researchers to replicate the study.

End of review.

Author Response

Thank you for providing a favorable review and comments for our manuscript.  Please find attached my responses to reviewer 1 comments highlighted in yellow.

Reviewer 2 Report

This study is well organized and written. No further suggestion can be given. However, this study may require a statistics section in the material and method, in order to descript the mean and SD with significances.  

Author Response

Thank you for providing a favorable review and comments for our manuscript. 

Response to Reviewer 2

Reviewer 2

This study is well organized and written. No further suggestion can be given. However, this study may require a statistics section in the material and method, in order to descript the mean and SD with significances.

COMMENTS: We thank reviewer 2 for their favorable comments.  We have included additional description of the statistical analysis performed in the Statistical analysis section.